# Trends of frequency, mortality and risk factors among patients admitted with stroke from 2017 to 2019 to the medical ward at Kilimanjaro Christian Medical Centre hospital: a retrospective observational study

Baraka Moshi [1,2] Nateiya Yongolo,[2,3] Sanjura Mandela Biswaro,[2,4] Hans Maro,[3,5] Sakanda Linus,[2,3] Stefan Siebert [6,7] William Nkenguye,[3] Emma McIntosh,[7] Febronia Shirima,[2,4] Rosalia E Njau,[2,4] Alice A Andongolile,[2,3] Manasseh Joel Mwanswila [2,4] Jo E B Halliday [8] Stefanie Krauth,[7,8] Kajiru Kilonzo,[3,4] Richard William Walker [9] Gloria August Temu,[3,4] Blandina T Mmbaga [2,3]

For numbered affiliations see end of article.

**Correspondence to**
Baraka Moshi;
bekamoshi0306@gmail.com

## ABSTRACT

**Objective** The burden of stroke has increased in recent years worldwide, particularly in low-income and middle-income countries. In this study we aim to determine the number of stroke admissions, and associated comorbidities, at a referral hospital in Northern Tanzania.

**Design** This was a retrospective observational study.

**Setting** The study was conducted at a tertiary referral hospital, Kilimanjaro Christian Medical Centre (KCMC), in the orthern zone of Tanzania.

**Participants** The study included adults aged 18 years and above, who were admitted to the medical wards from 2017 to 2019.

**Outcome** The primary outcome was the proportion of patients who had a stroke admitted in the medical ward at KCMC and the secondary outcome was clinical outcome such as mortality.

**Methods** We conducted a retrospective audit of medical records from 2017 to 2019 for adult patients admitted to the medical ward at KCMC. Data extracted included demographic characteristics, previous history of stroke and outcome of the admission. Factors associated with stroke were investigated using logistic regression.

**Results** Among 7976 patients admitted between 2017 and 2019, 972 (12.2%) were patients who had a stroke. Trends show an increase in patients admitted with stroke over the 3 years with 222, 292 and 458 in 2017, 2018 and 2019, respectively. Of the patients who had a stroke, 568 (58.4%) had hypertension while 167 (17.2%) had diabetes mellitus. The proportion of admitted stroke patients aged 18–45 years, increased from 2017 (n=28, 3.4%) to 2019 (n=40, 4.3%). The in-hospital mortality related to stroke was 229 (23.6%) among 972 patients who had a stroke and female patients had 50% higher odds of death as compared with male patients (OR:1.5; CI 1.30 to 1.80).

## STRENGTHS AND LIMITATIONS OF THIS STUDY

⇒ Our study contributes to the literature by providing crude estimates for stroke admissions, and co-occurrence with other non-communicable diseases.

⇒ This was a hospital-based study rather than a longitudinal community-based study, and thus may be subject to referral bias.

⇒ The data were collected retrospectively from written documents which are prone to have missing information due to reporting bias.

**Conclusion** The burden of stroke on individuals and health services is increasing over time, which reflects a lack of awareness on the cause of stroke and effective preventive measures. Prioritising interventions directed towards the reduction of non-communicable diseases and associated complications, such as stroke, is urgently needed.

## BACKGROUND

Stroke is defined by the WHO as an acute, focal or diffuse, dysfunction of the brain, originating from vessels and lasting for a period longer than a day.[1] Stroke is the second leading cause of death and the main contributing cause of disability in adults worldwide.[2,3] Stroke leads to multiple physical and cognitive impacts such as difficulty in communication, memory loss, challenges with mobility, depression and sometimes being bedridden.[4]

Globally, an estimated one in four adults will have a stroke during their lifetime. The burden of stroke in low-income

and middle-income countries (LMICs), including sub-Saharan Africa (SSA) has grown considerably in recent years.[3] The incidence of stroke in the past four decades (1970–2010) has increased by 100% in LMICs, but decreased by 42% in developed countries.[3 5] Most current reviews of stroke in patients, especially in young adults and older people, provide information about the aetiology and diagnostic approach written mainly from a high-income perspective and do not take global differences into account.[6 7]

Studies conducted in SSA have shown that stroke has been increasing over the years as evidenced by the observed proportions of admitted patients in tertiary hospitals. Interestingly, like most early hospital-based SSA studies, in a study from Ghana conducted over three decades there was a predominance of male admissions within the two earlier periods, but by the early 1990s, there were more women admitted to hospital with a stroke than men.[7] In a similar study from the Kilimanjaro Christian Medical Centre (KCMC) over four decades, there was a major increase in stroke admissions from the 1970s (less than two per year) to 2008 (more than 150 per year).[7] The changes made to primary healthcare in the last decade could be a major contributor to the change in hospital-seeking behaviour leading to a rise in tertiary hospital referrals.[8]

For most developed countries the incidence and mortality of stroke has been declining as a result of better hypertension control.[2 9] However, for LMICs, due to a lack of focused attention and limited resources directed to prevention and control of risk factors for stroke, its associated morbidity and mortality are steadily increasing.[10]

In Tanzania, incidence of stroke is well known, however efforts are still needed by policy makers to monitor the incidence, increase education and awareness, and develop better prevention interventions to reduce the number of stroke admissions. Identification of the burden of stroke will be crucial to increase general awareness of relevant risk factors along with stroke trends and their consequences.

Despite both global and national plans to reduce the burden of non-communicable diseases (NCDs), admission of patients who had a stroke is still rising. There is limited recent information about trends in stroke hospital admissions and outcomes among medical inpatients admitted to KCMC.

The specific objectives of this study are as follows:
1. To describe the sociodemographic characteristics, co-morbidities and their association with stroke among patients admitted at KCMC, Northern zonal referral hospital (2017–2019).
2. To determine the numbers of inpatients recorded with stroke for 2017, 2018 and 2019.
3. To determine outcomes of patients hospitalised with stroke at KCMC from 2017 to 2019.

## METHODS
### Setting
This study was conducted at KCMC, a zonal referral hospital within the Northern zone of Tanzania. The hospital serves as a referral hospital for over 15 million people in five regions, namely Kilimanjaro, Arusha, Tanga, Singida and Manyara. The healthcare system in Tanzania is a mix of both public and private services providers. The public healthcare system is organised into four levels: dispensaries, health centres, district hospitals and referral hospitals. Dispensaries and health centres provide basic healthcare services, while district and referral hospitals offer more specialised care. In terms of hospital organisation, Tanzanian hospitals, including KCMC, are typically organised into departments including internal medicine, general surgery, paediatrics, obstetrics and gynaecology, oncology, orthopaedics and others. The referral system goes from the dispensary level upward to the regional referral hospital, depending on the availability of required specialists and of medical equipment, including CT and MRI scanners. The management of hospitals is overseen by the Ministry of Health. The department of internal medicine specialises in prevention, diagnosis and non-surgical treatment. The medical ward is the unit/department where patients are admitted and treated for medical conditions that require non-surgical care. Patients who are treated or admitted to the medical ward include those with chronic illnesses such as diabetes, hypertension, heart disease, lung disease, kidney disease and liver disease, as well as those with infectious diseases.

Approximately 17 000 patients are seen at KCMC per month. The hospital has an in-patient capacity of 634 beds of which 100 are in the medical ward. The average daily admission rate for the medical department is 14 patients with a bed occupancy rate of 99% and an average length of stay of 8 days. The medical ward was the focus of this study because it is the main ward where patients who had a stroke who do not require surgical intervention are admitted.

### Study design
This was a retrospective observational survey, involving review of medical record files for individuals admitted to the medical ward at KCMC in the years 2017, 2018 and 2019.

### Study protocol
Patients' information was retrieved from hospital files stored in medical records and the electronic patient data registry. Files of patients admitted from 2017 to 2019 were retrieved by the medical records staff, and checked for eligibility. All eligible files were included for data extraction. Data were extracted using a prepared data extraction protocol, and entered into REDCap[11] software. Of the files 10% were double entered for quality control performed by the study coordinator.

Trained personnel captured information on a patient's age, gender, district of residence, level of education,

**Table 1** Sociodemographic characteristics of admitted patients who had a stroke (2017–2019) (N=972)

| Variable | Stroke admission | | | |
| | (2017–2019) | 2017 | 2018 | 2019 |
| | n=972 | n=222 | n=292 | n=458 |
|---|---|---|---|---|
| Age (years) | n (%) | n (%) | n (%) | n (%) |
| 18–45 | 94 (9.7) | 28 (12.6) | 26 (8.9) | 40 (4.7) |
| 46–60 | 205 (21.1) | 44 (19.8) | 65 (22.3) | 96 (21.0) |
| >60 | 673 (69.2) | 150 (67.6) | 201 (68.8) | 322 (70.3) |
| Mean (SD) | 57 (±32.6) | | | |
| Sex (n=971) | | | | |
| Male | 470 (48.4) | 101 (45.5) | 126 (43.2) | 243 (53.1) |
| Female | 501 (51.6) | 120 (54.1) | 166 (56.8) | 215 (46.9) |
| Marital status | n=538 | n=154 | n=199 | n=185 |
| Single | 33 (6.1) | 8 (5.2) | 17 (8.5) | 8 (4.3) |
| Married | 403 (74.9) | 115 (74.7) | 140 (70.4) | 148 (80.0) |
| Divorced | 16 (3.0) | 7 (4.5) | 6 (3.0) | 3 (1.6) |
| Widowed | 86 (16.0) | 24 (15.6) | 36 (18.1) | 26 (14.1) |
| Residence | n=966 | | | |
| Kilimanjaro | 825 (85.5) | 185 (83.3) | 249 (85.3) | 391 (85.4) |
| Arusha | 63 (6.5) | 17 (7.7) | 18 (6.2) | 28 (6.1) |
| Tanga | 10 (1.0) | 4 (1.8) | 2 (0.7) | 4 (0.9) |
| Manyara | 43 (4.5) | 4 (1.8) | 18 (6.2) | 21 (4.6) |
| Others | 25 (2.6) | 12 (5.4) | 5 (1.6) | 14 (3.1) |
| Source of hospital payment | | | | |
| Cash | 610 (62.8) | 148 (66.7) | 177 (60.6) | 285 (62.2) |
| Insurance | 341 (35.1) | 68 (30.6) | 110 (37.7) | 163 (35.6) |
| Social welfare | 7 (0.7) | 2 (0.9) | 1 (0.3) | 4 (0.9) |
| Not reported | 14 (1.4) | 4 (1.8) | 4 (1.4) | 6 (1.3) |

Among all identified patients who had a stroke, 229 (23.6%) died, while 458 (47.1%) stayed in hospital for more than 30 days (online supplemental table S1).

occupation, religion, tribe, diagnosis at the time of discharge, examinations and tests undertaken, duration of hospital stay, and number of admissions per year. Outcomes of admission such as death, referral to a higher facility and discharge were also collected.

The diagnosis captured was the final diagnosis recorded in the medical record at the time of discharge, death or referral.

## Study population
The study included adults aged 18 years and above, who were admitted to the medical wards from 2017 to 2019 with stroke, or having stroke during their stay as an inpatient.

## Data collection
Data were collected using a preformulated proforma which contained variables of interest to the study, as detailed above. A team of researchers obtained file numbers of all admissions to the medical ward for the years 2017 to 2019 from the medical records department. Using the file numbers obtained, team members retrieved the records from the hospital's medical records archive and electronic file systems. Data on demographic

characteristics, diagnoses, treatments and outcomes of admissions were documented anonymously in the proforma. Diagnoses and comorbidities were extracted from the final discharge diagnoses recorded in the files of the patients.

## Variable definitions
### Outcomes
The primary outcome was the proportion of patients who had a stroke admitted in the medical ward and the secondary outcome was outcome of admission.

## Secondary information
Sociodemographic characteristics of admitted stroke patients (age, gender, marital status, residence), source of hospital payment, discharge status, duration of hospital stay, diseases and associated conditions.

## Eligibility criteria
### Inclusion criteria
Adult patients aged 18 years and above, who were admitted to the medical wards from 2017 to 2019 with stroke, were included in the study.

**Table 2** Comorbidities and complications associated with stroke and its outcome in terms of inpatient mortality (N=972)

| Variable | Stroke | | Death |
| --- | --- | --- | --- |
| | Frequency | Percentage | n (%) |
| Hypertension | 568 | 58.4 | 130 (56.8) |
| Diabetes mellitus | 167 | 17.2 | 33 (14.4) |
| CCF | 46 | 4.7 | 14 (6.1) |
| Pneumonia | 241 | 24.8 | 110 (48.0) |
| CKD | 49 | 5.0 | 17 (7.4) |
| HIV/AIDS | 21 | 2.2. | 7 (3.1) |
| AKI | 7 | 0.7 | 4 (1.8) |
| Asthma | 3 | 0.3 | 3 (1.3) |
| COPD | 3 | 0.3 | 1 (0.4) |
| DKA | 7 | 0.7 | 3 (1.3) |
| UTI | 12 | 1.2 | 4 (1.8) |

AKI, Acute Kidney Injury; CCF, congestive cardiac failure; CKD, Chronic Kidney Disease; COPD, chronic obstructive pulmonary disease; DKA, Diabetic Ketoacidosis; UTI, Urinary Tract Infection.

### Exclusion criteria

All patients who were admitted directly to the intensive care unit and patients who died at the discharge point in emergency medicine were excluded from this study because their information were not captured in the database.

### Data analysis

The analysis was undertaken using STATA V.15 (StataCorp, College Station, Texas, USA). Categorical variables were summarised using frequencies and percentages, while numerical variables were summarised by the use of mean and SD. Proportional mortality was estimated by taking the total number of admissions divided by the total number of deaths for each diagnosed disease. Factors associated with stroke were estimated using multivariable logistic regression. All exposures that were found to have a significant association with the outcome (stroke), and those that were considered as possible confounders in univariable logistic regression, were entered for the development of the final model. A value of $p<0.05$ was considered statistically significant.

### Patient and public involvement

Patients and the public were not involved in the design and development of the study. The results of our study will be disseminated through open access publications, and publicised by other publicly accessible means.

### RESULTS
### Sociodemographic characteristics of patients who had a stroke

A total of 7976 patients were admitted from January 2017 to December 2019, of whom 972 (12.2%) were patients who had a stroke. The mean (SD) age of patients who had a stroke was 57 (±32.6) years. Ages of patients who had a stroke range from 24 years to 90 years. The majority (825, 84.5%) of the patients who had a stroke resided in Kilimanjaro region and 403 (41.3%) were married. Most (673, 69.2%) were aged 60 years and above, while 94 (9.7) were aged 45 years and below. More than half of the patients who had a stroke (501, 51.5%), were female. Moreover, the majority (610, 62.8%) of patients who had a stroke use cash for hospital payment (table 1).

### Comorbidities related to stroke, stroke rates and stroke mortality

Of the identified patients who had a stroke, 568 (58.4%) had hypertension while 167 (17.2%) had diabetes mellitus. Other comorbidities included congestive cardiac failure (46, 4.7%), chronic obstructive pulmonary disease/asthma (6, 1%) and HIV/AIDS (21, 2.2%). However, some patients who had a stroke had developed some complications whereby 241 (24.8) patients who had a stroke also had pneumonia and among them 110 (48%) died and 12 (1.2%) had Urinary Tract Infection (UTI) (table 2).

### Factors associated with stroke, as compared with factors associated with other admissions

Among those admitted, older people aged above 60 years had significantly six times higher odds of having a stroke as compared with the age group 18–45 years (OR: 6.0; 95 CI 4.90 to 7.63). Women had 20% higher odds of having strokes as compared with men (OR: 1.2; 95 CI 1.05 to 1.37).

After adjusting for other factors people aged above 60 years had four times higher odds of having stroke as compared with the age group 18–45 years (OR: 4.2; 95 CI 3.27 to 5.28). However married individuals had 60% higher odds of having a stroke as compared with the unmarried (OR:1.6; 95 CI 1.06 to 2.32) (table 3).

Among all identified patients who had a stroke 229 (23.6%) died while 458 (47.1%) stayed in hospital for more than 30 days (online supplemental table S1).

**Table 3** Factors associated with patients who had a stroke admitted at KCMC compared with inpatients without stroke

| Characteristics | Univariable | | | Multivariable | | |
|---|---|---|---|---|---|---|
| | OR | 95% CI | *p*-value | OR | 95% CI | *p*-value |
| Age (years) | | | | | | |
| 18–45 | 1 | | | 1 | | |
| 46–60 | 3.0 | 2.34 to 3.86 | <0.001 | 2.2 | 1.68 to 2.85 | <0.001 |
| >60 | 6.1 | 4.90 to 7.63 | <0.001 | 4.2 | 3.27 to 5.28 | <0.001 |
| Sex | | | | | | |
| Male | 1 | | | 1 | | |
| Female | 1.2 | 1.04 to 1.36 | 0.008 | 1.1 | 0.95 to 1.28 | 0.196 |
| Marital status | | | | | | |
| Single | 1 | | | 1 | | |
| Married | 3.5 | 2.49 to 5.15 | <0.001 | 1.6 | 1.06 to 2.32 | 0.024 |
| Divorced | 1.7 | 0.92 to 3.14 | 0.096 | 0.8 | 0.43 to 1.57 | 0.546 |
| Widowed | 5.0 | 3.30 to 7.63 | <0.001 | 1.6 | 1.04 to 2.61 | 0.034 |
| Level of education | | | | | | |
| No education | 1 | | | 1 | | |
| Primary | 0.5 | 0.21 to 1.04 | 0.066 | 0.9 | 0.36 to 2.09 | 0.749 |
| Secondary | 0.05 | 0.01 to 0.24 | <0.001 | 0.2 | 0.04 to 1.08 | 0.063 |
| Technical/university | 0.4 | 0.21 to 0.95 | 0.036 | 0.9 | 0.38 to 1.96 | 0.722 |
| Hypertension | | | | | | |
| No | 1 | | | 1 | | |
| Yes | 4.1 | 3.61 to 4.62 | <0.001 | 3.7 | 3.22 to 4.34 | <0.001 |
| DM | | | | | | |
| No | 1 | | | 1 | | |
| Yes | 1.1 | 0.90 to 1.29 | 0.405 | 0.9 | 0.82 to 1.08 | 0.391 |
| CCF | | | | | | |
| No | 1 | | | 1 | | |
| Yes | 0.3 | 0.19 to 0.36 | <0.001 | 0.2 | 0.11 to 0.21 | <0.001 |
| CKD | | | | | | |
| No | 1 | | | 1 | | |
| Yes | 0.4 | 0.28 to 0.50 | <0.001 | 0.3 | 0.21 to 0.39 | <0.001 |
| HIV | | | | | | |
| No | 1 | | | 1 | | |
| Yes | 0.2 | 0.15 to 0.37 | <0.001 | 0.4 | 0.27 to 0.67 | <0.001 |

CCF, congestive cardiac failure; DM, Diabetes Melitus; KCMC, Kilimanjaro Christian Medical Centre.

Trends show an increase in proportions of patients admitted with stroke compared with other conditions over the 3 years from 9.4% in 2017 to 11.9% in 2018 and 14.5% in 2019 (figure 1).

The number of patients who had a stroke increased over the 3 years consecutively from 222, 292 to 458, revealing that between 2018 and 2019 the numbers of patients who had a stroke almost doubled (online supplemental figure S1).

While the proportion of deaths of non-stroke patients decreased (35.2%, 34.6% and 30.2% in 2017, 2018 and 2019, respectively), the proportion of stroke deaths increased, with 11.9%, 16.7% and 19.8% for 2017, 2018 and 2019, respectively (figure 2).

From 2017 to 2019 the number of deaths for non-stroke patients decreased from 506, 498 to 435, respectively. However, for the same 3 years the number of deaths for patients who had a stroke increased from 60, 83 to 86 consecutively (online supplemental figure S2).

## DISCUSSION

This retrospective study looked at the number of adults with stroke diagnoses, out of all internal medicine (medical) admissions from 2017 to 2019 in KCMC, Tanzania. At 12.2%, the proportion of admissions with stroke is high as compared with the finding from the

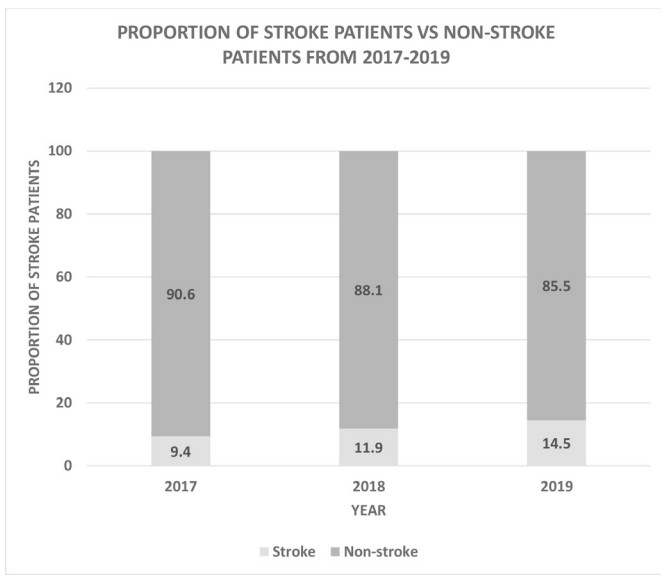

**Figure 1** Proportion of patients admitted with stroke compared with admission of non-stroke patients.

study in China looking on the prevalence and its related risk factors whereby the overall prevalence of stroke was 3.1%.[12] The burden of stroke and its impact on society is increasing and it has been recognised as a condition that needs urgent attention in LMICs. In this study we aimed to identify the burden of stroke and comorbid conditions in a consultant hospital. The proportion of overall stroke admissions for the 3 years was 22.8%, 30.0% and 47.0% for 2017, 2018 and 2019, respectively. This increase in the proportion of stroke admissions had doubled by the end of the study period. There were significant changes in the proportion of patients who had a stroke who died in hospital over the years of the study at 11.9%, 16.7% and 19.8%, respectively. The findings from our study had lower as compared to study conducted in Burkinafaso which had mortality rate of patients who had a stroke 39%, respectively, 37.6% in male and 41.6% in female

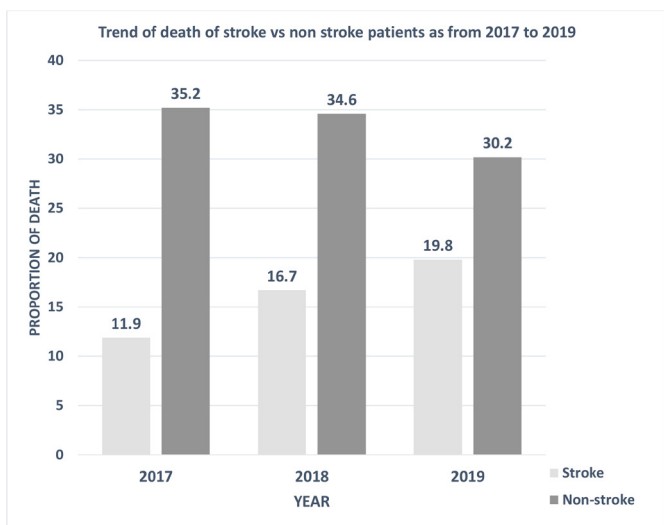

**Figure 2** Proportions of deaths of patients who had a stroke versus non-stroke patients (2017–2019).

where the mortality risk factors were the same. This disparity could be due to low access to health facilities for routine medical check-ups which could prevent the adverse outcomes of stroke and due to inadequate infrastructure and trained personnel with low level of care given in those established facilities in Burkina Faso.[13] We observed that the common NCDs related to stroke were hypertension and diabetes while the communicable disease was HIV/AIDS.

Over the 3 years of the study the stroke admission rate increased by 51.6%. This pattern of increasing stroke admissions over the years reflects a previous study conducted at KCMC comparing admission rates for stroke over the first four decades after KCMC opened in 1973 with less than two strokes per year in the 1970s increasing to over 150 stroke admissions per year in 2008.[7] The increase in stroke admissions over time could partly be explained by the increasing size of the population, increasing age of the population and potential changes in hospital usage. A previous study that included patients aged 60 years and over admitted to a hospital in Abeokuta, Nigeria, a hospital in Khartoum, Sudan and KCMC, over a 6-month period, showed that a quarter of the admissions at KCMC were due to stroke.[14]

A recently published study from Zanzibar looking at hospital admissions with stroke over a 1 year period found very high rates of stroke admissions, whereby, among 720 participants overall, 463 (64.3%) participants were experiencing a first-ever stroke, and 223 (31.0%) a recurrent stroke.[15]

The rate of in-hospital mortality in this study was 23.6% which was lower compared with the 27% reported in general medical inpatients admitted during the study period. The increased mortality of patients who had a stroke was associated with comorbidities such as hypertension 96 (41.9%), diabetes 26 (27.7%) and other NCDs. The case fatality rate was higher than the rate of 16.2% found in Ethiopia.[16]

In a European tertiary referral hospital study, mortality due to stroke decreased from 15.4% in 2008 to 9.8% in 2017.[8]

Mortality due to stroke in association with NCDs such as hypertension was high but the trend seems to be decreasing in our study through each year, 2017, 2018, 2019, respectively, which could be attributed to improved care and ease of access to medical services as well as an increase in access to health insurance services.

The most common risk factor for stroke identified in our study was hypertension, with a total prevalence of 24% among all admitted patients, and 58.4% of patients who had a stroke also had hypertension. This finding is consistent with other findings as uncontrolled hypertension is the most important risk factor for stroke, both in developing and developed countries. This trend may reflect poor community awareness, health practices and access to healthcare, including different patient-related factors. A community-based study in the Hai District, which is near KCMC, demonstrated very high rates of

undiagnosed hypertension in people aged 70 years and over, and that even in those who were diagnosed, relatively few were on drug treatment, and in those who were on drug treatment even fewer were well controlled.[7]

Diabetes mellitus is one of the major risk factors for the development of stroke. It was identified as a comorbidity in 167 (17.2%) patients. In our study the prevalence of DM was higher compared with a study by Boot *et al* which showed only 10% of patients who had a stroke had DM.[6] Ignorance of the risk factors and inability to manage such risk factors might be responsible for this effect.

Pneumoni is among the serious and common complication of stroke. Stroke-associated pneumonia is frequently caused by aspiration this is similar findings from our study revealed that the 24.8% among patients who had a stroke end up having pneumonia whereby, 110 (48%) died, the same to the findings of the study by Grossmann. Pneumonia occurs in 4%–10% of patients experiencing a stroke, making it one of the most problematic complications, associating it with a high mortality rate.[17]

Even when the patients understand the risk factors, they may not accept them to be the cause for stroke nor be able to afford the cost of medications. Additionally, since managing risk factors for stroke, such as hypertension, requires a longer period or may be lifelong, most patients fail to adhere and follow advice properly.[18] The above causes are likely to have contributed to the high rates of stroke among people with lower educational level, including rural farmers.

A previous community-based stroke incidence study from 2003 to 2006, which was conducted in the rural Hai District, near KCMC, and an urban area in Dar es Salaam, demonstrated very high stroke incidence rates, comparable to African-Americans in Manhattan, New York, USA.[19] This study involved active case findings and those who died before they were seen were picked up on verbal autopsy. Many patients do not go to hospital following stroke and there is usually a significant delay, so hospital statistics will underestimate the community burden.

## Study limitations

This was a hospital-based study rather than a longitudinal community-based study, may be subjected to referral bias whereby patients referred to KCMC may have some other complications which differ from patients treated to other health facilities. The type of data analysed in this study do not allow for a difference between risk factors, comorbidities and complications; as we have no data on the order of these factors, we present all factors as associated.

The data were collected retrospectively from written documents which are prone to have missing information subject to reporting bias.

## Strengths of the study

Our study contributes to the literature by providing crude estimates for stroke admissions, and co-occurrence with other NCDs.

## Conclusion

The burden of stroke on individuals and health services is increasing over time, which reflects a lack of awareness on both the modifiable and non-modifiable causes of stroke and effective preventive measures. Prioritising interventions directed towards the reduction of NCDs and associated complications such as stroke is urgently needed.

**Author affiliations**
[1]Epidemiology and Applied Biostatistics, Kilimanjaro Christian Medical University College, Moshi, United Republic of Tanzania
[2]Department of Clinical research, Kilimanjaro Clinical Research Institute, Moshi, United Republic of Tanzania
[3]Depatment of Epidemiology and Applied Biostatistics and Depatment of Internal Medicine, Kilimanjaro Christian Medical University College, Moshi, United Republic of Tanzania
[4]Department of Internal Medicine, Kilimanjaro Christian Medical Centre, Moshi, United Republic of Tanzania
[5]Department of Epidemiology, Tanzania Health Promotion Support (THPS), Moshi, United Republic of Tanzania
[6]Institute of Infection, Immunity and Inflammation, University of Glasgow, Glasgow, UK
[7]Depatment of Inflammation Medicine and Rheumatology, Health Economics and Health Technology Assessment (HEHTA), University of Glasgow Institute of Health and Wellbeing, Glasgow, UK
[8]School of Biodiversity, One Health and Veterinary Medicine, University of Glasgow, Glasgow, London, UK
[9]Depatment of One health and Veterinary Medicine, Newcastle University, Newcastle upon Tyne, UK

**Acknowledgements** The authors thank all participants who participated in one way of this work and the data collection team.

**Contributors** BM, BTM, NY, SB, RW, KK, SK and SL contributed to the initial draft, revisions and data analysis. and GT, BTB had the original idea for the research project, initiated the collaborative project and monitored data collection. BM, KK, NY, SK, SB, JH, EM, AA, WN, RW, MM, SS, FS, GT, RN, BTM, HM and SL designed the study, revised the paper and worked on the methodology component. BM is the guarantor.

**Funding** This work was supported by the National Institute for Health and Care Research (NIHR 17/63/35).

**Competing interests** None declared.

**Patient and public involvement** Patients and/or the public were not involved in the design, or conduct, or reporting, or dissemination plans of this research.

**Patient consent for publication** Not applicable.

**Ethics approval** This study involves human participants and was approved by the Ethical Committee of the Kilimanjaro Christian Medical University College Research Ethics and Review Committee (CRERC) in Moshi (KCMC/P. I/Vol.XI/2407); the National Institute of Medical Research Review Committee (NatHREC) of the National Institute Medical Research (NIMR) in Tanzania. (NIMR/HQ.R.8a/Vol.IX/3038); and the Medical Veterinary and Life Sciences (MVLS) ethics committee at the University of Glasgow (UofG), UK (200180100).This was a retrospective observational study that conducted a retrospective audit of medical record data from 2017 to 2019 for adult patients admitted with stroke or having a stroke while an inpatient, at the Kilimanjaro Christian Medical Centre.

**Provenance and peer review** Not commissioned; externally peer reviewed.

**Data availability statement** Data are available upon reasonable request. All data generated or analysed during this study are available from the Kilimanjaro Clinical Research Institute (KCRI) upon reasonable request from the corresponding author.

of the translations (including but not limited to local regulations, clinical guidelines, terminology, drug names and drug dosages), and is not responsible for any error and/or omissions arising from translation and adaptation or otherwise.

**ORCID iDs**
Baraka Moshi http://orcid.org/0000-0002-2894-5132
Stefan Siebert http://orcid.org/0000-0002-1802-7311
Manasseh Joel Mwanswila http://orcid.org/0000-0003-3378-2865
Jo E B Halliday http://orcid.org/0000-0002-1329-9035
Richard William Walker http://orcid.org/0000-0003-3155-122x
Blandina T Mmbaga http://orcid.org/0000-0002-5550-1916

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
