## [Reviewer comments · BMJ Open]

ARTICLE DETAILS

TITLE (PROVISIONAL)	Trends of frequency, mortality and risk factors among patients admitted with stroke from 2017 to 2019 to the medical ward at Kilimanjaro Christian Medical Centre hospital: a Retrospective Observational study
AUTHORS	Moshi, Baraka; Yongolo, Nateiya; Biswaro, Sanjura Mandela; Maro, Hans; Linus, Sakanda; Siebert, Stefan; Nkenguye, William; McIntosh, Emma; Shirima, Febronia; Njau, Rosalia E.; Andongolile, Alice A; Mwanswila, Manasseh; Halliday, J; Krauth, Stefanie; Kilonzo, Kajiru; Walker, Richard William; Temu, Gloria August; Mmbaga, Blandina

VERSION 1 – REVIEW

REVIEWER	Sobolewski, Piotr Collegium Mecicum UJK
REVIEW RETURNED	10-Feb-2023

GENERAL COMMENTS	Understanding the epidemiology of stroke and stroke risk factors in low-income countries is of great interest to many clinicians. This is how I view this scientific report. However, the average reader does not understand the health care system, the organization of the hospital in a country like Tanzania. Explain to the reader what a “medical word” is. For me, it is not clear who is treated in the medical ward and who in other departments of the hospital. In this context, I don't understand the sentence: The hospital has an in-patient capacity of 634 beds of which 100 are in the medical ward. Another term internal (medical) medicine. The more so that the authors excluded: All patient under Intensive care unit. A description of the structure of the hospital is necessary. It is also necessary to expand all abbreviations in the text and in the table descriptions.
---

REVIEWER	Saylor, Deanna Johns Hopkins University, Neurology
REVIEW RETURNED	15-Feb-2023

GENERAL COMMENTS	Moshi, et al present the results of a retrospective review of admissions for stroke to the adult medical wards at Kilimanjaro Christian Medical Center in Arusha, Tanzania. While these results are interesting, the introduction does not clearly justify why this study was conducted and how it adds to previously conducted studies on similar topics from the region (and even from KCMC itself). Furthermore, discrepancies in reported data from different sections of the manuscript and inaccuracies in tables raises concern about the data analysis and reported results as described in more detail below.
--

INTRODUCTION:

1.) The last sentences of the 3rd and 4th paragraphs is identical.
2.) The authors seem to undermine the purpose of their study in the introduction by stating definitively that stroke admissions are increasing over time and that females are beginning to make up the preponderance of stroke admissions in sub-Saharan Africa, making their stated purpose - to determine if stroke admissions are increasing - and one of their findings - that females make up the majority of the population - less compelling. Furthermore, it seems that some of this work and these findings have already come from their institution. Please clarify in this section why the presented work is still important and unique.

SPECIFIC OBJECTIVES

1.) It seems objectives 1 and 3 are very similar and could be combined. Also, generally these are included in the last paragraph of the introduction rather than a separate manuscript section.

METHODS

1.) This section can be rewritten in paragraph form with fewer subsection as is typical for a manuscript.
2.) Please justify why patients admitted to the ICU were excluded as it seems this may have biased the study to omit the most severe strokes.

RESULTS

1.) I am unclear why pneumonia and UTI are included in comorbidities along with other comorbidities that are stroke risk factors. It seems these are likely post-stroke/in-hospital complications rather than associated with stroke occurrence.
2.) Page 15/Table 3: The abstract states that in-hospital mortality was 23.6% while this section states it is 15.9%. Please clarify.
3. Page 25/Table 3: Please double check these numbers as they do not make sense. Table 3 and the text states that the majority of stroke patients (n=458) had a length of stay >30 days, but the percentage listed is 12.8%. Furthermore, there are ~7000 non-stroke patients, and the table states 3117 had a length of stay >30 days. This should be less than half of the non-stroke patients, but the percentage listed is 87.2%. Please clarify.
4.) The abstract states that between 2% and 6% of annual admissions were due to stroke, but the results state that the overall prevalence of stroke was 12.2% (last sentence on page 13). Please clarify how this is possible.
5.) It is somewhat misleading to state that the "proportion of overall stroke admissions" was 22, 30 and 47% over the three years. The wording suggests you are comparing the proportion of all admissions that were stroke admissions, which seems the most straightforward way to present these data since this manuscript is looking at whether the burden of stroke admissions in the medicine department is increasing. Rather, you are reporting the proportion of all stroke admissions that occurred in each year. This proportion could be very misleading. For example, what if the total number of admissions to the medicine wards doubled in one year? If that was the case, you could see a huge increase in the proportion of total stroke admissions that occurred that year without actually seeing an increasing proportion of the admissions of the medicine wards due to stroke.

DISCUSSION:

	1.) I am still unclear why the association with pneumonia is being interpreted as a potential causal mechanism for stroke as it seems it is in the last sentence of the 1st paragraph. Isn't it more likely that pneumonia occurs after the stroke due to aspiration? 2.) In general, new results should not be initially presented in the Discussion. From my reading of the results, there is no report that older patients were more likely to have multiple comorbidities. Please present this data in the results section. 3.) Again, in the discussion, in-hospital mortality rate is reported as 23.5% but is reported as 15% in the results. Please clarify. 4.) The discussion states hypertension was present in 24% of participants but the results state it was present in 58%. Please clarify. 5.) In the Discussion, it is important to try to put the results of other studies that you report in the context of your results. In many parts of this discussion, it reads as literature review. It would significantly strengthen the discussion if you report the results of another study to then put those results in the context of your own findings. WERE they similar to or different than this study's findings? If different, why might these differences be? It is also important to give context to reported results. For example, lines 18 and 19 on page 17 quote a study with a diabetes prevalence of 10% but the location of this study is not mentioned and is crucial to putting these results in context. 6.) It is important to add a discussion of the strengths and limitations of the paper. 7.) A conclusion paragraph is usually added at the end of the Discussion. 8.) It is not standard manuscript format to include an "extras" section. This is an interesting fact that should be incorporated and contextualized within the rest of the Discussion section. MINOR: 1.) The "key messages" section could use editing for grammatical clarity.
--	--

VERSION 1 – AUTHOR RESPONSE

Reviewer comments	Action	Page number and line number
Please remove the 'Key message' section.	We thank the reviewer for bringing this to our attention, hence removed.	Page 4
Please ensure that you have fully discussed the methodological limitations of the study in the Discussion section of the main text.	We have added methodological limitations in discussion section.	Page 18, section-discussion the line 12-14.
What is the health care system, the organization of the hospital in a country like Tanzania?	We have added an explanation of the healthcare system and organization in Tanzania.	Study setting section, Page 6, first nine lines.

What is medical ward?	We thank the reviewer for bringing this, Medical ward is a hospital unit or department where patients are admitted and treated for medical conditions that require non-surgical care. We have added a respective definition to the manuscript.	Study setting section, Page 6, line 10-11.
Who is treated in the medical ward and who in other departments of the hospital?	Patients who are treated or admitted to the medical ward include those with chronic illnesses such as diabetes, hypertension, heart disease, lung disease, kidney disease, and liver disease. They may also include patients with infectious diseases. A sentence explaining which patients are treated in the medical ward has been added to the study setting section.	Study setting section, Page 6, line 12-13.
The hospital has an in-patient capacity of 634 beds of which 100 are in the medical ward (Explanation).	The entirety of the hospital is capable of admitting 634 patients at full capacity within which 100 belong to the internal medicine department also referred to as medical wards. The explanation has been added to the paper.	Study setting section page 7, first three lines.
What is internal medicine?	Thank you, reviewer, for rising this, the description of the term is given as requested.	Study setting section, Page 6, line 9.
The reason for excluding all patient under Intensive care unit.	We excluded patients who were admitted straight to ICU and who either died there or were discharged from there but have included those who were admitted to a medical ward and subsequently transferred to ICU. Very few individuals will have been directly admitted to ICU and missing information of the variables of interest.	Eligibility criteria section, Page 9, first two lines.
A description of the structure of the hospital.	KCMC is organized into departments incl. internal medicine, general surgery,	Study setting section, Page 6, last five lines, and first five lines in Page 7.

	orthopedic, pediatrics, obstetrics and gynecology and cancer unit.	
It is also necessary to expand all abbreviations in the text and in the table descriptions.	List of abbreviations expanded.	Page 2.
Problem statement and justification	We have added an explanation of the problem and the rationale.	Page 5, Introduction section, line 8-11
The last sentences of the 3rd and 4th paragraphs are identical.	We thank the reviewer for bringing this to our attention and have removed one of the sentences.	Page 5, Introduction Section.
Please clarify in this section why the presented work is still important and unique.	We thank the reviewer for bringing this. In Tanzania the previous stroke incidence is known but still efforts are needed by policymakers to monitor the incidence and primary prevention is needed to reduce the number of stroke admissions as there are very limited recent data.	Page 5, Introduction section, line 8-12
It seems objectives 1 and 3 are very similar and could be combined	The objectives have been updated.	Page 5, Objective section, last five lines
Objective one and three are included in the last paragraph of the introduction rather than a separate manuscript section.	We agree with the reviewer and now the objectives have been removed and added to the objective section where they should be.	Introduction section, Page 5
The methods section can be rewritten in paragraph form with fewer subsection as is typical for a manuscript.	Changes made as requested.	Method section, page 6-9.
Please justify why patients admitted to the ICU were excluded as it seems this may have biased the study to omit the most severe strokes.	We included patients who were admitted to a medical ward first and then transferred to ICU, and excluded patients who were admitted straight to ICU and either died there, or were discharged from there. Very few individuals will have been directly admitted to ICU and missing information of the variables of interest.	Page 10, exclusion criteria section first three lines.

Why pneumonia and UTI are included in comorbidities along with other comorbidities that are stroke risk factors. It seems these are likely post-stroke/in-hospital complications rather than associated with stroke occurrence.	We agree that pneumonia and UTI are complications of stroke rather than risk factors. Hence these have been edited.	Page 12, Result section (Table 2). Page 16, discussion section, line 19-21.
The abstract states that in-hospital mortality was 23.6% while this section states it is 15.9%. Please clarify.	23.6% is the overall mortality for all hospital admissions while 15.9% is mortality for only stroke patients. This has been clarified.	Page 3, abstract result section and page 13 (Table 3).
Please double check these numbers as they do not make sense. Table 3 and the text states that the majority of stroke patients (n=458) had a length of stay >30 days, but the percentage listed is 12.8%. Furthermore, there are ~7000 non-stroke patients, and the table states 3117 had a length of stay >30 days. This should be less than half of the non-stroke patients, but the percentage listed is 87.2%. Please clarify.	We agree with the reviewer and have hence corrected the percentage in table 3 accordingly. At first, it was the column percentage.	Page 13, result section (Table-3)
The abstract states that between 2% and 6% of annual admissions were due to stroke, but the results state that the overall prevalence of stroke was 12.2%	2.8%, 3.6% and 5.7% are stroke prevalence for the year 2017, 2018 and 2019 respectively. And 12.2% is overall prevalence for the entire three years.	Page 3-Abstract section.
It is somewhat misleading to state that the "proportion of overall stroke admissions" was 22, 30 and 47% over the three years. The wording suggests you are comparing the proportion of all admissions that were stroke admissions, which seems the most straightforward way to present these data since this manuscript is looking at whether the burden of stroke admissions in the medicine department is increasing. Rather, you are reporting the proportion of all stroke admissions that occurred in each year. This proportion could be very misleading. For example, what if the total number of admissions to the medicine wards doubled in one year? If that was the case, you could see a huge increase in the proportion of total stroke admissions that occurred that year without actually seeing an increasing	We agree with the reviewer the percentage was calculated considering the stroke as denominator, but in this case taking the overall admission as the denominator will make sense. So, in that case the proportions will read as 9.4%, 11.9%, 14.5%.	Discussion section, Page 16, last seven lines. Also see figure 4.

proportion of the admissions to the medicine wards due to stroke.		
Explanation on the discussion section on why the association with pneumonia is being interpreted as a potential causal mechanism for stroke as it seems it is in the last sentence of the 1st paragraph. Isn't it more likely that pneumonia occurs after the stroke due to aspiration?	We agree that pneumonia and UTI are complications of stroke rather than risk factors. Hence these have been edited.	Page 12, result section. (Table 2).
In general, new results should not be initially presented in the Discussion. From my reading of the results, there is no report that older patients were more likely to have multiple comorbidities. Please present this data in the results section.	We agree with the reviewer that the results need to be initially presented in the results section and have made the relevant changes.	Page 17, discussion section. Page 10, Result section, line 7-8.
In the discussion, in-hospital mortality rate is reported as 23.6% but is reported as 15% in the results. Please clarify.	23.6% is the overall mortality for all hospital admissions while 15.9% is the mortality for only stroke patients.	Page 16, discussion section. The fourth line from end. Page 13, result section, Table 3.
The discussion states hypertension was present in 24% of participants but the results state it was present in 58%. Please clarify.	The 58% includes patients with both Hypertension and Stroke while the 24% relate to patients with hypertension among all hospital admissions.	Page 17. Result section, page 12, result section Table 2.
In the Discussion, it is important to try to put the results of other studies that you report in the context of your results. (Restructure the discussion section)	The discussion has been restructured, by involving what others has done in this area.	Page 17, discussion section, last 8 lines
It is important to add a discussion of the strengths and limitations of the paper.	We agree with the reviewer and have hence added a strength and limitations description in the discussion section.	Page 18, section-discussion, line 3-8.
A conclusion paragraph is usually added at the end of the Discussion. It is not standard manuscript format to include an "extras" section. This is an interesting fact that should be incorporated and contextualized within the rest of the Discussion section.	We agree with the reviewer and the conclusion paragraph has been included at the end of the discussion section.	Page 18, section-discussion, line 10-13
	In addition to the reviewers' directly requested changes, we	

	have added some additional clarifications and language changes to improve the overall comprehensibility and language of the manuscript. We have highlighted these changes in the “changes highlighted” document.	
--	--	--

VERSION 2 – REVIEW

REVIEWER	Saylor, Deanna Johns Hopkins University, Neurology
REVIEW RETURNED	17-Jun-2023

GENERAL COMMENTS	The authors have satisfactorily responded to my initial concerns about the manuscript. The statistics, in particular, are much improved and, I believe, presented in a more meaningful and interpretable way. I believe this manuscript is an important addition to the literature.
---

VERSION 2 – AUTHOR RESPONSE